# A Combined Methods of Senile Trees Inventory in Sustainable Urban Greenery Management on the Example of the City of Sandomierz (Poland)

Wojciech Durlak [1], Margot Dudkiewicz [2],* and Małgorzata Milecka [2]

[1] Horticultural Production Institute, Faculty of Horticulture and Landscape Architecture, University of Life Sciences in Lublin, 28 Głęboka St., 20-612 Lublin, Poland

[2] Department of Landscape Architecture, Faculty of Horticulture and Landscape Architecture, University of Life Sciences in Lublin, 28 Głęboka St., 20-612 Lublin, Poland

* Correspondence: margot.dudkiewicz@up.lublin.pl; Tel.: +48-81-531-96-70

**Abstract:** The sustainable management of urban greenery consists, among others, of the inventory, valuation, and protection of trees of monumental size. This article presents the results of the inspection of 13 large trees growing in the city of Sandomierz, located in south-eastern Poland. The examined specimens belong to five species: Norway maple (*Acer platanoides* L.), common ash (*Fraxinus excelsior* L.), white poplar (*Populus alba* L.), English oak (*Quercus robur* L.), and small-leaved lime (*Tilia cordata* Mill.). The health condition of the trees was assessed using acoustic and electrical tomography, as well as chlorophyll fluorescence tests. Diagnostics employing sound waves and electrical resistivity were crucial in assessing tree health. The data based on chlorophyll fluorescence confirmed the results obtained during tomographic examinations. It was an innovative combination of three non-invasive methods of examining the health condition of trees and their valuation. Economic valuation allows us to reduce to common denominator issues that are often difficult to decide due to different perspectives—expressing the economic value of trees. Calculating the value of trees allowed us to show the city's inhabitants the value of trees that are of monumental size. Thanks to the cooperation of scientists with the city authorities, an economic plan for trees of monumental size was created, distinguished by an individualized and holistic approach to each specimen covered by the study. The database prepared has a chance to become an effective management instrument used by environmental protection authorities and a source of knowledge and education for the city's inhabitants.

**Keywords:** sustainable management; urban greenery; senile trees; monumental trees; Picus Sonic Tomograph 3; Picus TreeTronic; fluorometer; Sandomierz; Poland

## 1. Introduction

Finding a balance between social, ecological, and economic aspects is one of the most significant challenges in sustainable city management [1–3]. An important problem is managing and shaping properly functioning urban green areas with the preservation of trees of monumental size. The most important management procedure in taking care of trees of monumental size, which are commonly referred to as veteran trees as they usually are old as well, is to think of how to improve their living conditions, e.g., by improving and protecting their rooting area, or gradually improving their light acquisition [4,5]. Senile and veteran trees should be given adequate space; they need enough light and room for the trunk and crown and ample below-ground rooting and soil volume. They should have the opportunity to continue to grow old and be left alone where possible, in favorable surroundings. Sometimes, the care of trees of monumental size includes pruning to remove dead branches, cutting crowns to reduce their weight and volume, and as bracing and other safeguards. Monitoring and control should be carried out not only through visual

assessment (VTA) but also with the use of modern technologies. Likewise, protecting historic trees includes entering them into the register of natural monuments [6,7].

Hazardous trees regularly lead to the death or injury of arborists and property owners. It is worth mentioning in this context that the ISA has created the Tree Risk Assessment Qualification (TRAQ). TRAQ promotes the safety of people and property by providing a standardized and systematic process for assessing tree risk. The results of a tree risk assessment can provide tree owners and risk managers with information to help make informed decisions to enhance tree benefits, health, and longevity [8].

This study applies to an important theme in the urban environment, the management of senile trees, which are very valuable for biodiversity, human well-being, and cultural history. The objective of this study was to formulate recommendations for senile trees based on the results of their condition assessment obtained with the use of three non-invasive methods. Studying the condition of the trees is the first step of making a management plan to preserve these trees safely in the environment. The manuscript studied different methods for evaluating the condition of the trees and a method for calculating the tree replacement value.

*Need to Protect Older Trees in Cities*

In difficult urban conditions, a significant number of trees die or are removed before they reach full maturity [9,10]. When trees grow old, they are considered to be 'senile'. The decline of trees starts with a sparse appearance, yellowing and other types of foliage symptoms, undergrowth, and sickly appearance and dried-up top growth. The branches of trees start to die from the top downwards. External features include the spiral grain in a tree's trunk, thin or balding bark, the loss of apical dominance, crown dieback, and crowns with a few, large limbs, among others [11–13]. Pederson described six external characteristics of potentially old angiosperm trees. These characteristics are found in the stems and crowns of trees and include: (1) smooth bark; (2) low stem taper; (3) high stem sinuosity; (4) crowns comprised of few, thick, twisting limbs; (5) low crown volume; and (6) a low ratio of leaf area to trunk volume. Admittedly, characteristics (4) and (5) are often correlated; a crown with low volume often has only a few twisting limbs. The greater the number of these characteristics in one tree, the greater the likelihood that the tree is old [14].

Senile trees are precious for a place's nature, landscape, culture, and tradition. Unfortunately, more and more often, it turns out that, due to the rapidly progressing urbanization of the landscape, they grow in collision with the urban infrastructure. As a result of which, they are often mechanically damaged or deformed as a result of improperly carried out works by municipal services dealing with infrastructure, and, as a result, they enter the dieback phase too early with a distinct dryness in the crown. Here, it should be emphasized that, most often, measures of crown dieback or chlorosis are used. As a result, they are vulnerable to disposal due to the safety risk to people and property. Old trees, however, are significant witnesses to the history of towns and villages [14,15]. They are a practical component of the cultural landscape—they are often painted, photographed, and described. Moreover, they are also an essential element of the natural world. They are the habitat of rich biological and microbial life. According to British data, over 2000 species of invertebrates (6% of British invertebrate fauna) depend on the habitats of senile trees; therefore, the removal of such trees seriously affects the biodiversity of towns and villages [16,17].

Urban green spaces regulate regional microclimate via shading, evapotranspiration, boosting air movements, and increasing heat exchange, which mitigates urban heat island (UHI) effects at a city scale [18–20]. Furthermore, vegetation in green spaces can not only intercept rainfall and reduce rainfall runoff, but it also brings more rainfall infiltration, which decreases the frequency of urban floods and stormwater treatment costs and damages [21]. Most studies at regional or city scales show a modest modeled reduction in pollution concentration of less than 5% resulting from urban vegetation [22,23]. Trees increase both the surface roughness (slowing air flow thus enhancing deposition and absorption pollutant removal processes) and the area of the ground surface that atmospheric pollutants come into

contact with (acting as biological filters, enhanced by the properties of their surfaces) [24]. The introduction of trees within a street canyon also has the potential to significantly alter the soundscape by generating sounds associated with the rustling of leaves in response to wind and attracting bird wildlife sounds that would be rated more positively than a street canyon dominated by road traffic noise [25]. Moreover, trees growing in cities are an element that humanizes space [26].

Several aspects of physical health have been shown to be correlated with aspects of urban "greenery", such as mortality, longevity, heart rates, and weight changes [27,28]. There are also numerous studies relating aspects of mental health to the prevalence of vegetation [29,30]. Responses of human health to actual and perceived biodiversity have been generally categorized as those that cause or reduce harm and that restore or build the capacity for physical and mental health [31]. Some urban residents are highly deprived of virtually any access to nature, such that even modest additions have been shown to have measurable positive effects [32,33]. However, it should be noted that different people may have divergent views. The owners of the property may have an entirely different opinion regarding the same tree, seeing only the inconvenience in the development and maintenance of the property caused by its shading by the crown of the tree or falling leaves or appreciating the beauty of the specimen and its environmental value. Still, other qualities are noticed by someone who looks at a given tree from the street or a window of an apartment in a nearby building and sees it, above all, as an indispensable and permanent element of space, a sentimental landscape. Such a subjective perception of a tree can be related to its value, understood in social, psychological, and economic categories [34–36]. Thanks to such a view of nature, it is easier to reach the imagination of, for example, politicians or city authorities, or, it may be the only way most people will realize that a functioning environment is more valuable than a destroyed one. It is also worth making society aware that the landscape we live in is about ourselves and our culture.

Sustainable landscape management needs to take into account both the natural values of the landscape (e.g., trees) and the human values. The combination of visual inspection (VTA), acoustic and electrical tomography, and stress testing with a fluorometer is a practical approach to measuring the health of various tree species. Thanks to the use of sound tomography, the threats can be quickly diagnosed and restoration work can be carried out. However, the decision-making process should be accompanied by a process of social participation. Social awareness of the value of trees is needed to manage senile trees in the city space and to support various decisions of officials and arborists (e.g., entering into the register of monuments or tree maintenance procedure).

## 2. Materials and Methods

The research presented in this article shows a combination of several non-invasive methods for detecting damage and cavities inside trunks and the resistance of the trunk to the fracture of city trees. The combination of visual inspection (VTA), acoustic tomography, electrical tomography, and stress testing with a fluorometer is an effective instrument to show the health status of various tree species. It is worth noting here that the evaluation of the statics and condition of trees are two different but important issues. The developed procedures enable the analysis of endangered trees and can improve the risk assessment process near the tree or the risk caused by it. In the presented study, the economic value of trees was also estimated.

The study covered 13 trees (12 nature trees of monumental size, and one tree is a candidate tree) growing in Sandomierz, belonging to five species: white poplar (*Populus alba* L.)(4), pedunculate oak (*Quercus robur* L.) (4), small-leaved lime (*Tilia cordata* Mill.) (3), ash(*Fraxinus excelsior* L.) (1 pc.), and Norway maple (*Acer platanoides* L.)(1 pc.).The trees were selected for research due to the fact that they are all natural monuments of the city of Sandomierz. They grow in various spatial situations: parks, sacred areas, school areas, and as street greenery. The research was carried out in 2021. The study used quantitative and qualitative data processing methods. The following numerical parameters

characterizing the tested trees were determined: circumference at the height of 130 cm (cm), the range of the crown (m), and the height of the tree (m). The crown's reach was measured in two directions, N-S and E-W, and the obtained results were averaged. The height of the trees was measured with a Nikon Forestry Pro laser rangefinder. The location of trees was determined based on GPS positioning. When describing the trees, particular attention was paid to the health condition of the trunk (possible necrotic strips, cavities, rot, traces of insect feeding, mushroom fruiting bodies, trunk tilt) and crowns (drought, broken branches, asymmetry) and to the prevailing habitat conditions. The condition of the immediate vicinity of the tree was documented, such as the type of surface, the proximity of buildings, and the intensity of pedestrian and vehicle traffic. The historical and cultural valuesand the role of trees in a location were assessed, e.g., certain individual trees have significance in relation to specific persons or historical events, or some specimens of trees are remnants of old gardens and parks that no longer exist.

*2.1. Sandomierz—General Information and Location of Research Tress*

Sandomierz is a historic city in south-eastern Poland, situated on the Vistula River, with 23,000 residents. Numerous tourist routes run through the city, including the Cistercian route, St. Jakub, Via Jagiellonica, Wooden Architecture, and the Eastern Bicycle Trail "Green Velo." There are many monuments in Sandomierz, and the Old Town is a well-known historical urban, architectural, and landscape complex.

In the area of Sandomierz, there are 12 specimens of old trees under protection in the form of nature monuments (Figure 1). In Poland, one of the formal attributes of a monument tree is the circumference. The trees assessed grow in different parts of the city (Figure 2). There are three natural monuments in Piszczele Park, one Norway maple and two white poplars. The Norway maple (*Acer platanoides* L.) (1) grows in high densities on a slope among other trees on Emmendingen Avenue. Two magnificent white poplars (*Populus alba* L.) located near the main pedestrian and bicycle path leading through the park are also monumental (2 and 3). In Saski Park, the largest in Sandomierz, located near the Opatowska Gate, on the side of J. Słowacki Street, there is a monumental pedunculate oak (*Quercus robur* L.) (4). The remaining trees of monumental sizeare located near historic sacred buildings, e.g., the old small-leaved lime (*Tilia cordata* Mill.) grows at the entrance to the Dominicans' property from the side of St. James, in the corner of the brick fence (5). The second of the limes (6) is located at the very entrance to the St. James church. On the other hand, at the bell tower of the church of St. Paul the Apostle, the common ash (*Fraxinus excelsior* L.) grows (7), and in the vicinity of the presbytery buildings, there is an imposing English oak (*Quercus robur* L.) (8). Within the Old Town of Sandomierz, there is another natural monument, a small-leaved lime (*Tilia cordata* Mill.) (9) growing near the Collegium Gostomianum building (now High School No. 1 in Sandomierz) at J. Długosz Street. However, at Milbert Street, a magnificent pedunculate oak (*Quercus robur* L.) grows (10). The following two monuments of nature are the white poplars (*Populus alba* L.) (11 and 12) growing in the right-bank part of Sandomierz on the playground at K. K. Baczyński Street. A specimen that meets the conditions of a massive tree was also studied, an English oak (*Quercus robur* L.) (13) from Opatowska Street, which was proposed for protection as a natural monument.

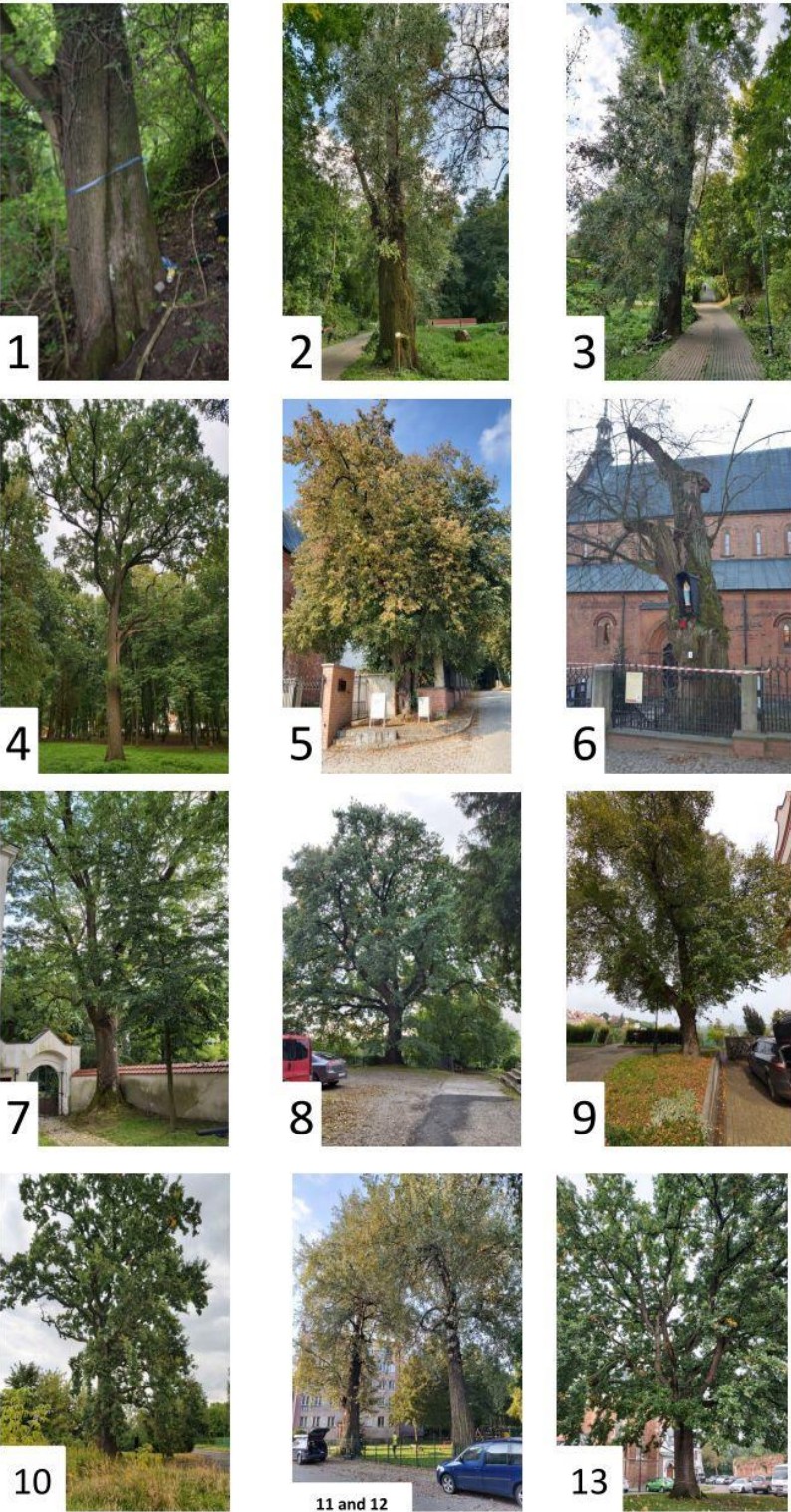

**Figure 1.** Trees of monumental size in Sandomierz. (**1**) *Acer platanoides* in Park Piszczele; (**2**) and (**3**) two trees of the species *Populus alba* in Park Piszczele; (**4**) *Quercus robur* in Saski Park, (**5**) and (**6**) two trees of the *Tilia cordata* species at the church of St. James; (**7**) *Fraxinus excelsior* at the church of St. Paul; (**8**) *Quercus robur* at the rectory of the Church of St. Paul; (**9**) *Tilia cordata* at Collegium Gostomianum; (**10**) *Quercus robur* at Milbert Street; (**11,12**) two *Populus alba* trees at Baczynski Street; (**13**) *Quercus robur*, acandidate for a natural monument, Opatowska Street (photo by M. Dudkiewicz, 2021).

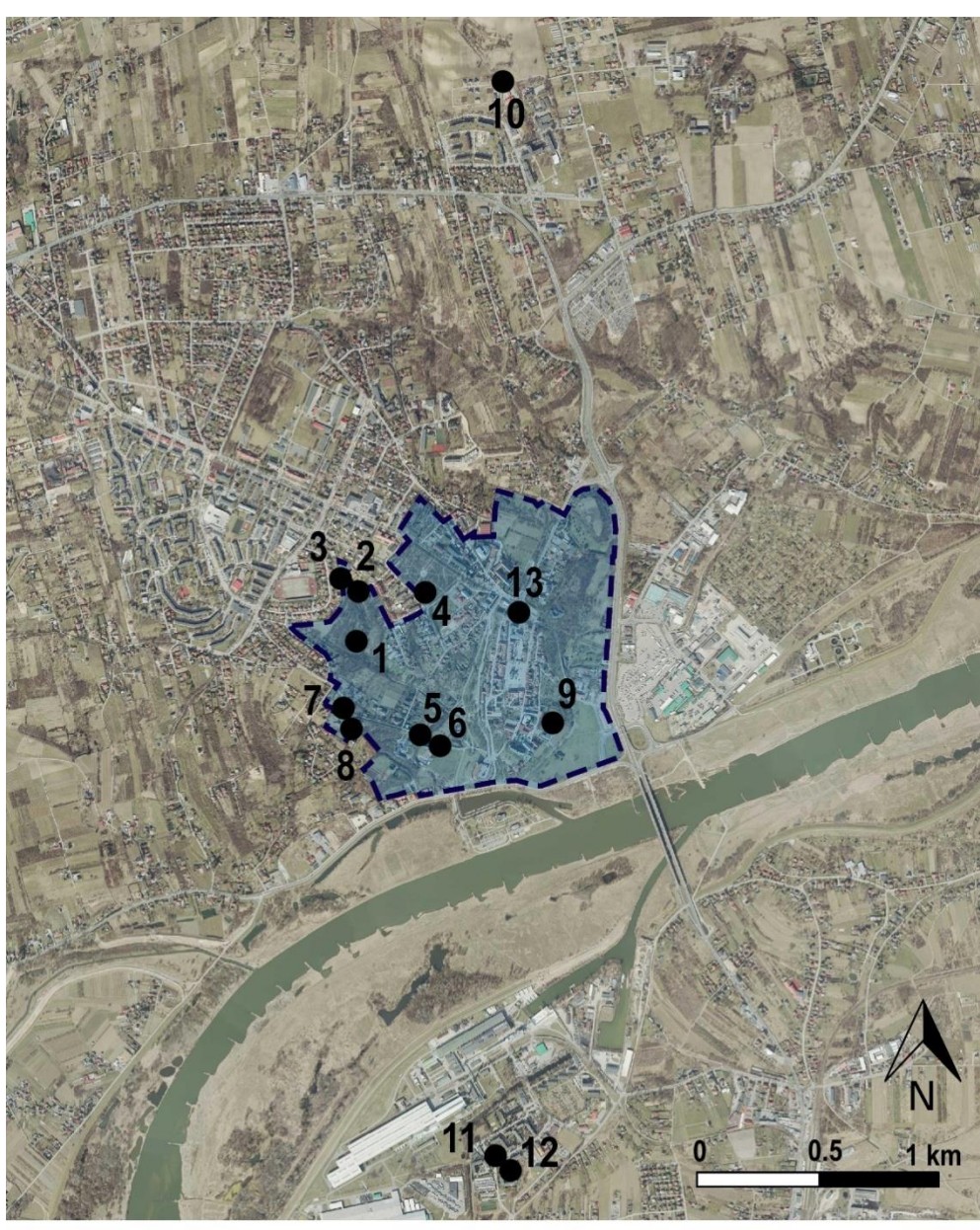

**Figure 2.** Location of trees of monumental size in the city of Sandomierz (made by M.Milecka).

*2.2. Measures*

2.2.1. Tree Age Assessment

Several methods were used to calculate the age of trees: age tables by Majdecki [37]; the standard formula for assessing the age of trees (trunk circumference 130 cm/2.5 = tree age); age tables of Mydłowska [38]; and a tree age calculator [39]. It should be noted that the age of the trees is visually difficult to estimate. The significant emerging discrepancies in assessment result mainly from different calculation methods. Therefore, it is best to consider similar results as the most reliable. Trees from the same planting period may often differ in the circumference of the trunks. This state of affairs can be influenced by many factors, such as habitat conditions, soil fertility, soil moisture, growth rate, or genetic factors.

### 2.2.2. Acoustic Tomography

In the case of old trees, diebacks are the leading cause of their fall or breakage. Structurally weak specimens are more susceptible to strong wind or heavy snow. As weather anomalies resulting from climate change occur more frequently, the number of incidents and financial damage caused by falling trees in cities is increasing. Internal flaws are difficult to assess by visual inspection alone, which makes managing and preparing for the appropriate risks difficult [40]. The method of risk assessment VTA—based on the visual assessment of an expert—is often insufficient to provide a reasonable opinion [41–43].

Generally, two types of methods are used to diagnose internal defects in tree trunks or branches: invasive and non-invasive. The first one involves the use of devices that interfere with the internal tissues of the tree, which include *Presler auger*. The second, however, does not damage the wood structures and uses acoustic or electromagnetic waves. In the case of old trees, especially those designated as protected trees, the physical damage to trees should be minimized, and non-invasive diagnostic methods should be used. Therefore, diagnostic examinations of tree trunk health conditions were carried out using the Picus Caliper and Picus Tomograph (sonic and electric) of the German company Argus Electronic GmbH (Figures 3 and 4).

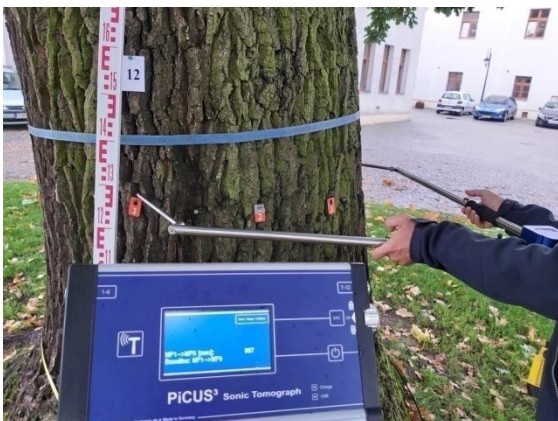

**Figure 3.** Stem geometry measurement with the electronic PiCUS Caliper (photo by M. Dudkiewicz, 2021).

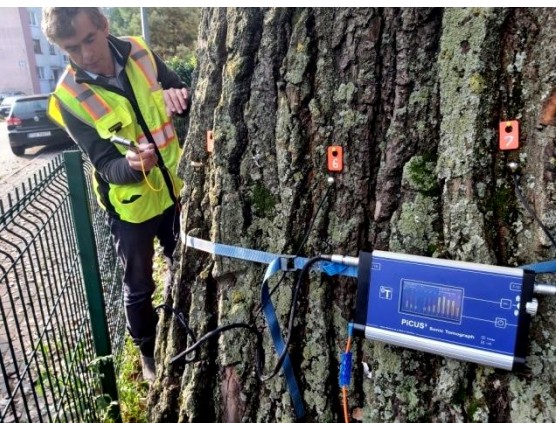

**Figure 4.** PiCUS sonic tomograph central unit and tree trunk measurement (photo by M. Dudkiewicz, 2021).

Acoustic tomography enables the non-invasive measurement of the internal structures of a tree trunk in terms of detecting rot or decay using sound waves, without the need to make drills that are harmful to the tree [44,45]. It is worth remarking that a tomograph study considers the condition of wood only at the very point of cross section of the measurement,

not any higher or upper (of course, it can be used to conclude the condition of the trunk in general).Acoustic tomography consists of a central unit, sensors placed around the tree trunk and connected with pins driven shallowly into the trunk, and specialized software. Tomographic examinations were carried out at a height of 50 to 160 cm above the ground, depending on the tree. The optimal test height was determined on the basis of tapping the trunk with a rubber mallet at different levels beforehand. Then, after interpreting the appearing noises, a decision was made on the level of distribution of the measurement sensors on the tree trunk. Sensors measure the time of the sound wave propagation in the wood caused by the impact of an electronic hammer. The distances between the sensors are measured using a special PiCUS Caliper operating in a Bluetooth system, which allows for a detailed mapping of the shape of the tree trunk. The obtained and processed results are a colored tomogram showing the changes inside the tree trunk. At this point, it is worth noting that a tomogram taken at a single time shows the conditions at the time of measurement. Determining a change in stem characteristics requires repeated measurements over time. When analyzing the measurement results, particular attention should be paid to the color of the obtained image, which determines the so-called wood density map. Different colors mean different speeds of sound propagation inside the trunk, depending on the elasticity and density of the wood. The light brown to black color corresponds to the range of the speed of sound from 60 to 100% of the highest speed of sound, which means a living and healthy wood tissue. Various shades of green color correspond to the speed of the sound wave from 40 to 60% of the speed of sound at an average level, which is equivalent to a slight deterioration of the structure of the wood. The color pink means the propagation of sound from 20 to 40%, and a color from blue to white is in the range of 0–20% (the slowest speed of sound). Therefore, these are the areas with the weakest structure, where there is damage and intense decay of wood. The yellow lines on the cross-section of the trunk suggest the appearance of internal cracks. Internal cracks are not automatically dangerous. It depends on their location and the structure and condition of the tree/trunk/branch [46,47].

To obtain more precise results, sonic tomography with electrical tomography is suggested. As both research methods complement each other, the result is a detailed image of the inside of the tree trunk [48,49].

### 2.2.3. Electric Tomography

The PiCUS TreeTronic (Figure 5) is a device that uses an electrical voltage to define areas within a tree trunk of different electrical resistivity. The result of such a study is a two-dimensional map of the impedance of the wood called the electrical resistivity tomogram. The electrical resistivity of wood depends on various factors. These include the tissue's water content, the concentration of elements (ions), or the structure of cells (the so-called reaction wood has a different resistivity than regular wood). As in the case of sonic tomography, the obtained tomogram is in the form of a colored map of the trunk's cross-section, on which the marked colors correspond to a specific value of electrical resistivity. Blue indicates low-resistivity areas characterized by high water content in the tissues. Green and yellow indicate increased resistivity, and red corresponds to high resistivity (low water content, etc.). To analyze the electrical resistivity tomogram, three types of resistivity in trees are considered: Type 1, Type 2, and Type 3.

Type 1 is the low resistivity (high conductivity = blue colors) of the areas facing outwards and the high resistivity (lower conductivity = red colors) of the areas towards the inside of the tree trunk. Each species has a typical wet and dry distribution. Most European tree species, i.e., maple, chestnut, birch, chestnut, beech, ash, poplar, willow, rowan, linden, oak, elm, larch, pine, spruce, and many others, are classified as Type 1. Such trees usually have lower sapwood (blue) resistivity on the edges and higher resistivity in heartwood (red). Chestnut trees or poplars often develop inner wet wood, even when young. Wet wood changes the resistivity of the heartwood. However, it should be considered that the detected early stages of wet wood do not threaten the trunk's stability. Type 2 is high

resistivity (low conductivity = red) to the outside and low resistivity (high conductivity = blue) towards the inside of the tree trunk. Type 3 shows a ring system, where high resistivity (low conductivity = red) covers the middle part of the trunk and low resistivity (high conductivity = blue) is located in the area facing the outside of the trunk and its core part. It is a typical arrangement for healthy trees, e.g., English oak [50].

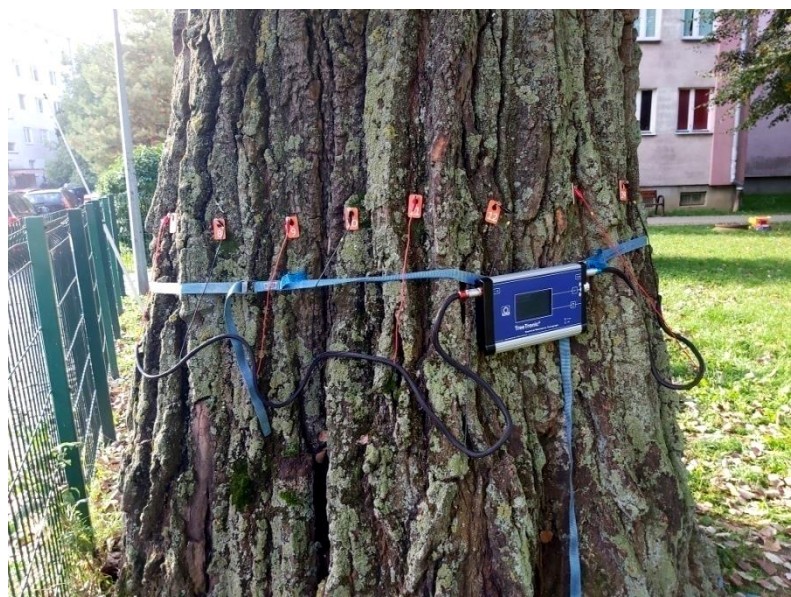

**Figure 5.** Tree measurement with the Picus TreeTronic electric tomograph (photo by M. Dudkiewicz, 2021).

2.2.4. Measure the Photosynthesis Process

Using the OS5p+ fluorometer produced by the American company Opti Sciences, the stress level in the examined trees was measured, which allowed for assessing their condition. The measurement of chlorophyll fluorescence and the analysis of this phenomenon is a recognized method that allows you to quickly and non-invasively check the photosynthesis process and determine the health condition and the occurrence of stress phenomena in plants. Clips with a blackout tab were placed on the leaves in different parts of the crown of each tree. After 30 min, chlorophyll fluorescence was measured.

The fluorometer is a device that allows quick and non-invasive measurements of the photosynthesis process (Figures 6–8). The measurement of chlorophyll fluorescence is particularly useful for assessing the health condition of plants and in situations where various environmental factors affect them [51–53]. The clips were attached, 5 pieces per tree, at equal intervals around the crown, at crown base height (CBH).The measurement procedure is based on placing a unique clip on the leaf blade, closed for about 20–30 min (dark time needed for the measurement). Then, the measurement is made by inserting the probe tip into the clip and opening the window. The display shows the results defining the level of the respective coefficients. One of them ($F_V/F_M$ determinant, maximum photochemical efficiency) is commonly used as a reliable diagnostic indicator, among others, to evaluate the impact of stressful conditions on plants [54]. In conditions of total growth and the absence of stress factors in plants, this index, depending on the species, ranges from about 0.83 to 0.85 relative units [55]. The action of biotic and abiotic stressors reduces the quantum efficiency of the photosystem II (PSII) and its value. Very low indices of this parameter—at the level of 0.20–0.30—indicate the occurrence of irreversible changes in the PSII structure [56]. It should be pointed out that fluorescence measurements mostly consider the functioning of the photosynthetic system (especially PSII) in the leaves and indicate the effect of some stresses but not all, e.g., it does not consider the condition of the trunk or branches, i.e., about the structural stability at all. There may be internal decay in the trunk, but the outer layers of conductive tissues are perfectly functioning and the

photosynthetic system is working fine, and although the chlorophyll measurements are not good, this should not be a cause for concern.

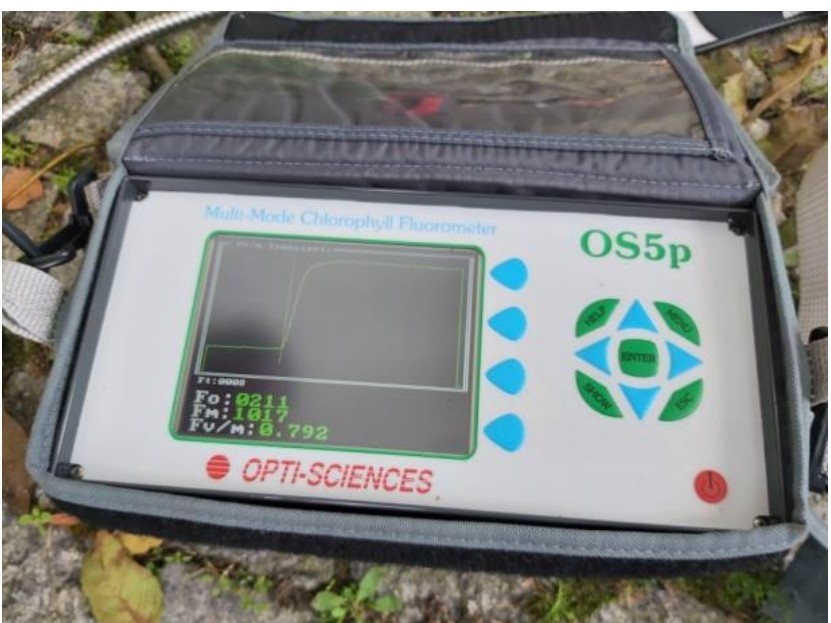

**Figure 6.** Advanced Fluorometer OS5p+ to measure the photosynthesis process (photo by M. Dudkiewicz, 2021).

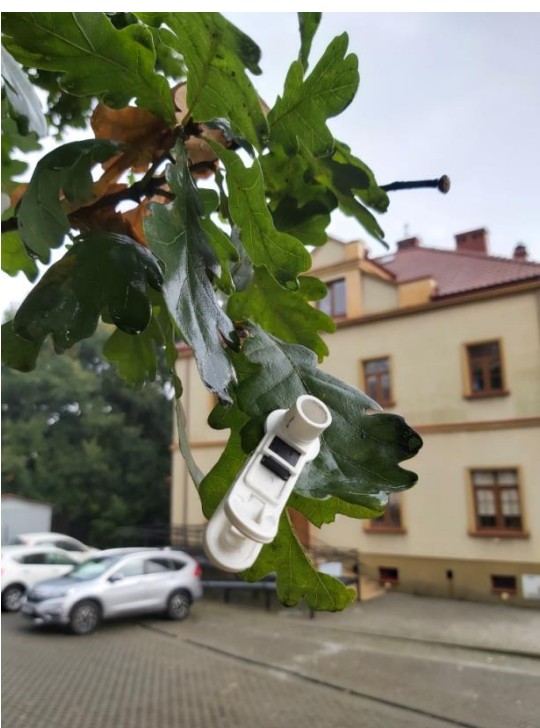

**Figure 7.** Measurement clip (photo by M. Dudkiewicz, 2021).

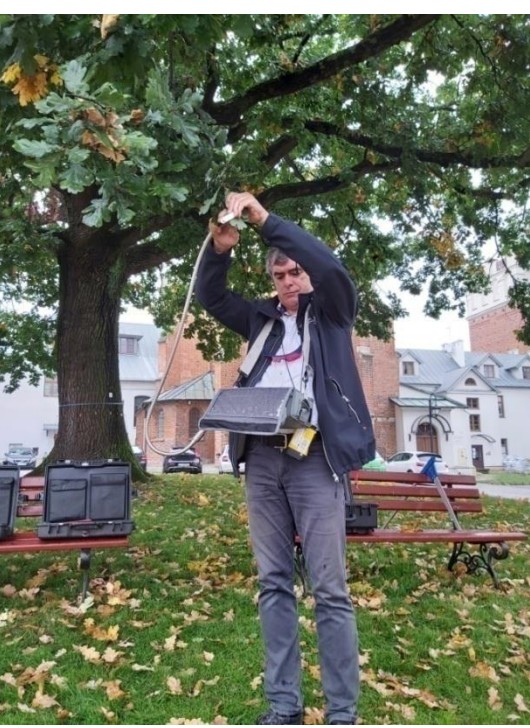

**Figure 8.** The moment of taking the measurement (photo by M. Dudkiewicz, 2021).

2.2.5. Economic Valuation of Trees

Economic valuation methods are used around the world to give an economic value to trees. Many regions and countries in Europe have produced their own economic valuation model, e.g., CAVAT in the UK, VAT03 in Denmark, Koch's method in Germany, and Norma Grenada in Spain, and the same trend can be seen outside of Europe, such as with CTLA in the USA and the Revised Burnley Method in Australia. The valuation of the value of trees was performed based on the method developed by the team of authors led by Szczepanowska [57,58]. In this case, the tree valuation method used in Poland is based on comparative analyses of foreign methods, including in the EU countries, and on domestic analyses. It uses American solutions promoted by the CTLA to a large extent, taking into account the conditions prevailing in Poland. However, it should also be pointed out that this method does not take into consideration the value of the ecosystem services of the tree nor the biodiversity effect, but it is purely a tree replacement cost. Thus, in the case of veteran trees with high biodiversity value, it unfortunately underestimates the actual value of the tree.

The method assumes that growing trees constitute a permanent value (this is confirmed by the content of Article 48 of the Civil Code, which states that trees constitute a component of the land from the moment of planting or sowing). The basis for the valuation of trees in the method is the cost of tree replacement. The method distinguishes between the following two situations: when a damaged tree can be physically replaced by a comparable tree available on the market (then the basis of the valuation are the costs of a tree analogous to a growing tree) and when the tree cannot be physically restored (in such a case, the replacement cost is considered as the basis for the valuation of trees available as nursery material, additionally enlarged using increment factors). The coefficients are determined for a given tree size, considering the tree's species value, growth rate, and adaptation to urban conditions.

When assessing the value of the stand, the following formula was used:

$$RWD_{>25} = W \times G \times P \times K \times L$$

where:

RWD$_{>25}$—actual value of a tree with a trunk circumference of more than 25 cm;

W—base value of the tree;

G—tree species value (tabular list of species, the value of the tree is higher for e.g., oak and lower for poplars);

P—tree growth rate coefficient (tabular list of species);

K—tree health factor;

L—tree location factor [58–60].

The obtained results concerning the actual value of trees are presented in Table 1.

**Table 1.** Results of the dendrological survey of trees of monumental size in Sandomierz (by authors).

| No. | Species Name | The Circumference of the Trunk at the Height of 1.3 m (cm) | Height (m) | Crown Reach (m) | Localization | Age | Mean Age | Value (Euro) * |
|---|---|---|---|---|---|---|---|---|
| 1 | Norway maple (*Acer platanoides* L.) | 275 | 13.2 | EW 18.6m NS 15 m Mean: 16.8 m | Piszczele Park 50°40′56.9″ N 21°44′40.1″ E | 157 [1] 110 [2] 157 [3] 158 [4] | 145 | 6 820.50 |
| 2 | White poplar (*Populus alba* L.) | 577 | 36.6 | EW 18.8 m NS 23.7 m Mean: 21.2 m | Piszczele Park 50°40′57.8″ N 21°44′25.0″ E | 152 [1] 230 [2] 152 [3] 260 [4] | 198 | 11 674.03 |
| 3 | White poplar (*Populus alba* L.) | 472 | 35 | EW 22.2 m NS 24 m Mean: 23.1 m | Piszczele Park 50°40′57.6″ N 21°44′26.0″ E | 124 [1] 188 [2] 124 [3] 213 [4] | 162 | 8 284.30 |
| 4 | English oak (*Quercus robur* L.) | 397 | 17.6 | EW 25.5 m NS 24.5 m Mean: 25 m | Saski Park 50°40′56.3″ N 21°44′38.1″ E | 275 [1] 158 [2] 275 [3] 313 [4] | 255 | 21 723.34 |
| 5 | Small-leaved linden (*Tilia cordata* Mill.) | 390 | 10.8 | EW 12.8 m NS 11.7 m Mean: 12.2 m | At the entrance to the vineyard and the Church of St. James 50°40′37.1″ N 21°44′41.0″ E | 162 [1] 156 [2] 162 [3] 268 [4] | 187 | 13 536.18 |
| 6 | Small-leaved linden (*Tilia cordata* Mill.) | 384 | 8.4 | EW 10 m NS 9 m Mean: 9.5 m | At the entrance to the Church of St. James 50°40′37.8″ N 21°44′39.3″ E | 159 [1] 153 [2] 159 [3] 264 [4] | 184 | 9 475.32 |
| 7 | Common ash (*Fraxinus excelsior* L.) | 310 | 22.2 | EW 21 m NS 21 m Mean: 21 m | At the Church of St. Paul 50°40′57.6″ N 21°44′26.0″ E | 165 [1] 124 [2] 164 [3] 175 [4] | 157 | 16 547.62 |
| 8 | English oak (*Quercus robur* L.) | 525 | 18.4 | EW 27 m NS 28 m Mean: 27.5 m | At the rectory of the Church of St. Paul 50°40′40.2″ N 21°44′22.9″ E | 364 [1] 210 [2] 364 [3] 414 [4] | 338 | 49 692.52 |
| 9 | Small-leaved linden (*Tilia cordata* Mill.) | 419 | 13.8 | EW 14.2 m NS 18.9 m Mean: 33.1 m | At the Collegium Gostomianum 50°40′39.2″ N 21°45′6.2″ E | 174 [1] 167 [2] 174 [3] 288 [4] | 201 | 16 976.20 |
| 10 | English oak (*Quercus robur* L.) | 532 | 18.4 | EW 20.7 m NS 18.5 m Mean: 19.6 m | Milberta St. 50°42′3.8″ N 21°45′0.4″ E | 369 [1] 212 [2] 369 [3] 384 [4] | 333 | 23 378.06 |
| 11 | White poplar (*Populus alba* L.) | 465 | 18.6 | EW 17.7 m NS 17 m Mean: 17.3 m | Baczyńskiego St. 50°39′41.4″ N 21°44′51.9″ E | 122 [1] 186 [2] 123 [3] 210 [4] | 160 | 12 051.42 |
| 12 | White poplar (*Populus alba* L.) | 430 | 17.8 | EW 18.3 m NS 15 m Mean: 16.6 m | Baczyńskiego St. 50°39′41.1″ N 21°44′52.2″ E | 113 [1] 172 [2] 113 [3] 194 [4] | 148 | 10 804.72 |
| 13 | English oak (*Quercus robur* L.) | 357 | 16.4 | EW 25.3 m NS 26.5 m Mean: 25.9 m | Opatowska St. 50°40′51.6″ N 21°45′0.1″ E | 247 [1] 142 [2] 247 [3] 281 [4] | 229 | 44 574.54 |

* converted from PLN to EUR at the rate of 1 Euro = 4.70 PLN of 20.07.2022.; [1] according to Majdecki's age table; [2] according to the formula, circumference/2.5 = age of the tree; [3] according to Mydłowska's tables; [4] http://www.tree-guide.com/tree-age-calculator (accessed on 26 October 2022).

### 3. Results

*3.1. Trees and Examination*

The first step in the research was to assess each tree to visualize signs of structural instability, mechanical damage, or fungi and prepare the necessary photographic documentation.

The results of the dendrometric tests are presented in Table 1.

*3.2. Results of Computed Tomography*

Since chemical properties change earlier than physical properties, combining acoustic and electrical tomography enabled a more advanced assessment of the internal tree trunk structures. In numerous instances, the measurement result can be used to identify the type of rot or determine whether the inside of the trunk is moldy or infected with bacteria. In addition, the result of the electrical test also provides information on the degree of moisture in the wood. A low impedance value indicates a high water content. The distribution of areas with different resistivity (different humidity) can also be used to analyze the efficiency of the root system and water transport inside the tree.

To find out more about these results and to further explain them or create a clearer image, the collected data are presented in the Supplementary File S1. Table 2 summarizes the results of tomographic examinations in the form of generated tomograms.

**Table 2.** Results of the computed tomographyof senile trees in Sandomierz(by authors).

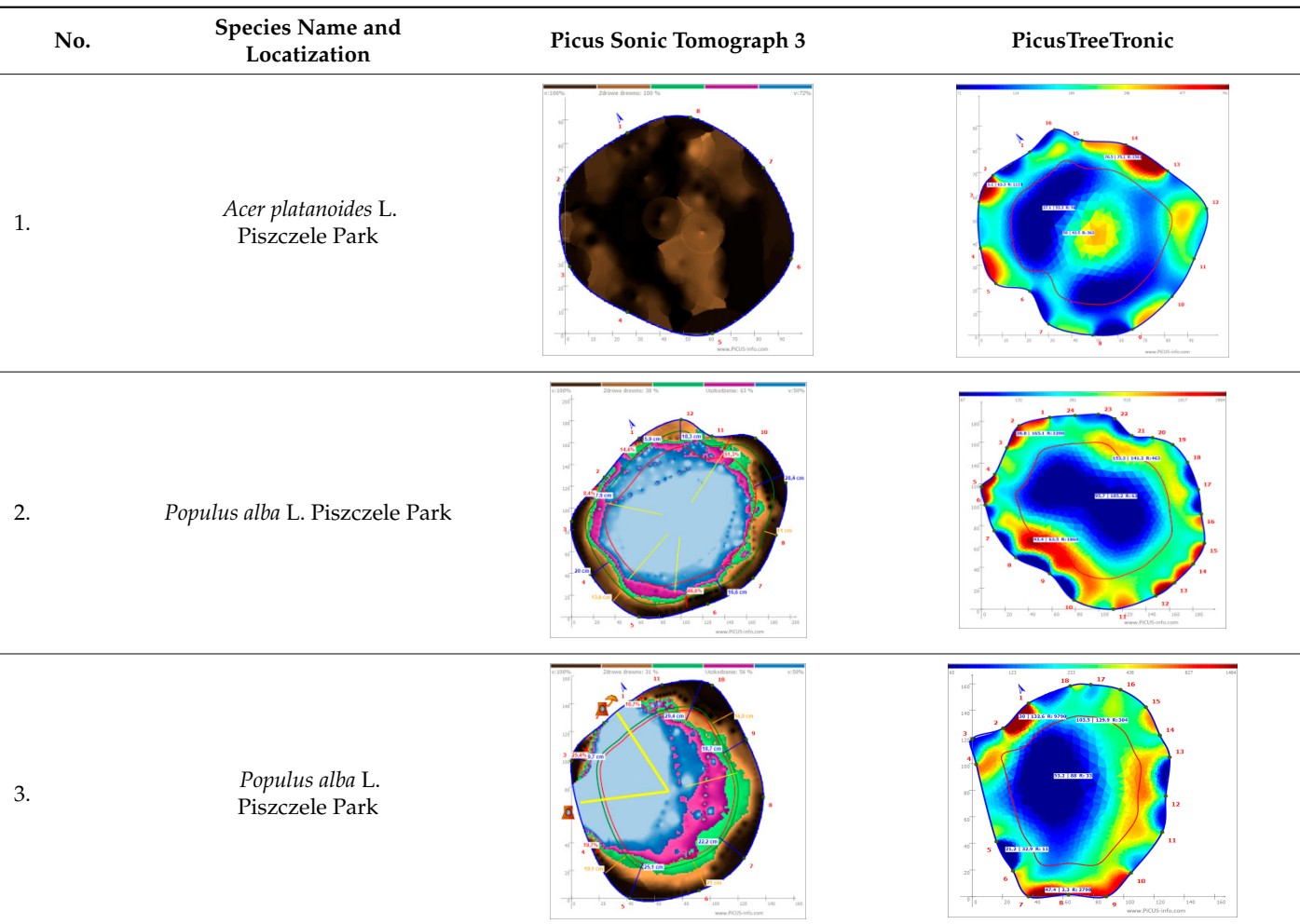

| No. | Species Name and Localization | Picus Sonic Tomograph 3 | PicusTreeTronic |
|---|---|---|---|
| 1. | *Acer platanoides* L. Piszczele Park | | |
| 2. | *Populus alba* L. Piszczele Park | | |
| 3. | *Populus alba* L. Piszczele Park | | |

**Table 2.** *Cont.*

| No. | Species Name and Localization | Picus Sonic Tomograph 3 | PicusTreeTronic |
|-----|-------------------------------|-------------------------|-----------------|
| 4. | *Quercus robur* L. Saski Park |  |  |
| 5. | *Tilia cordata* Mill. At the entrance to the vineyard and the Church of St. James |  |  |
| 6. | *Tilia cordata* Mill. At the entrance to the Church of St. James |  |  |
| 7. | *Fraxinus excelsior* L. At the church of St. Paul |  |  |
| 8. | *Quercus robur* L. At the rectory of the Church of St. Paul |  |  |

**Table 2.** *Cont.*

| No. | Species Name and Locatization | Picus Sonic Tomograph 3 | PicusTreeTronic |
|---|---|---|---|

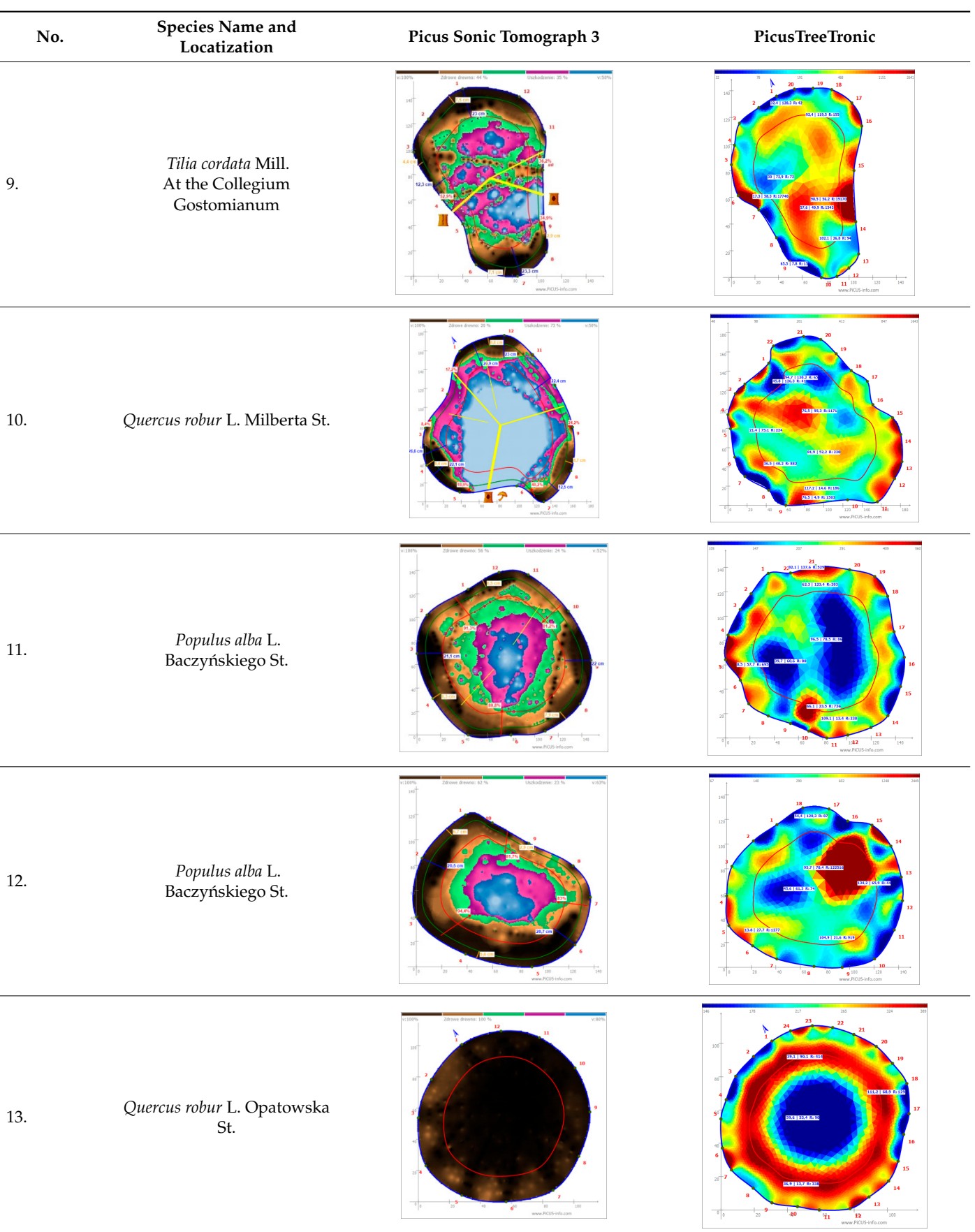

| No. | Species Name and Locatization |
|---|---|
| 9. | *Tilia cordata* Mill. At the Collegium Gostomianum |
| 10. | *Quercus robur* L. Milberta St. |
| 11. | *Populus alba* L. Baczyńskiego St. |
| 12. | *Populus alba* L. Baczyńskiego St. |
| 13. | *Quercus robur* L. Opatowska St. |

The Effect of Hollowness on the Statics of Senile Trees

Scientists have developed a large body of literature about tree biomechanics. In the 1960s, arborists began to describe trees as engineered structures, using equations and terms such as modulus of rupture, applied bending moment, and lever arm. Biomechanical experiments are designed to quantify the forces imposed on trees and their ability to support the load or fail [61,62].

A hollowing trunk is a natural process, and it is not necessarily a sign of an ailing tree. The center of the tree is deadwood, which is slowly decayed by fungi. Hollow tubes have the advantage of resisting bending and torsional moments with a relatively lower weight per unit length than solid cylinders of the same weight [63]. These hollow structures are ubiquitous in nature, such as bamboo stems, cereal stalks, decayed hollow tree trunks [64], and animal bones [65], as well as in artificial structures, such as buildings, bridge frames, athletic halls, and aircraft fuselages. Hollow-bearing trees are being removed at a faster rate than hollows form in many parts of the world (urban and rural) [66]. There are ongoing declines in the abundance of fauna dependent on hollow trees for survival. Nowadays, many hollow trees are cut down from fear that they may fall at some point in the future and injure someone. Yet, many hollow trees are clearly structurally sound, as they remain in place for decades. Tree trunks can be up to 70% hollow before the probability of failure suddenly increases [67].

The safety assessment of hollow trees has always fascinated arborists, and the criteria to be employed have led to severe public discussions in the professional scene in Europe. Based on Mattheck et al. (1995), many tree consultants state that the required thickness of the residual wall should not be below a t/R ratio of 0.3 to prevent shell-buckling and cross-sectional flattening [68]. On the other hand, Wessolly and Erb (1998) published an opposite theory, by which much lower thickness are often calculated and accepted [69]. Their methods (SIA and Elastomethod) are based on the bending theory of the hollow beam. The degree of hollowness or different wood strengths were only two quite subordinate aspects of the statics triangle, which consists of load, shape, and material. The size of the actual cavity provides no information on the safety of the tree; even a safe tree may be nearly entirely hollow. There is no fixed boundary value such as 0.3, for example. The importance of the cavity can only be assessed in comparison with the static cushion of the tree. This requires a load analysis, simplified by DIN 1055 with the appropriate cw-value or, even better, matched to trees by a static integrated tree assessment (SIA) [70].

Sometimes, strong wind can significantly affect a tree's stability. Wind–tree interactions take place at a wide range of temporal and spatial scales [71]. Aerodynamic drag at all surfaces of the aerial parts of trees—from individual leaves to whole tree crowns [72]—perturbs the airflow inside forest canopies [73]. Wind load acts on the tree crown and causes large bending moments on the trunk and root plate. These moments are the main sources of trunk failure and root overturning. The breaking of thick-walled hollow tree trunks subjected to gale gusts at the crown usually results from bending stresses. Failure begins with fiber buckling on the compression side, followed by fiber tearing on the tension side. It has been indicated that tangential cracking due to cross-sectional flattening, followed by longitudinal splitting, is dominant for hollow trunks. Growth stresses are ubiquitous in trees and may affect the failure mechanism. The longitudinal growth stress of normal wood is tensile on the surface of the trunk and is compressive inside [74], whereas tangential growth stresses are always compressive on the surface but tensile inside. Because the longitudinal compression strength of wood is only about one-third of its tensile strength, longitudinal tensile growth stresses can partially compensate for the compressive stresses on the leeward side of a trunk induced by the bending moment of wind loads acting on the crown. The compressive surface growth stress in the tangential direction can partially resist longitudinal splitting [75]. Furthermore, as the branches move around in the wind, they dissipate wind energy, which reduces the load transferred to the trunk and increases the mechanical stability of the tree [67]. These features may be recognized as the self-optimized design of trees as a result of evolution.

Old hollow trees with severe mechanical faults and old lapsed pollards that are prone to structural collapse require priority treatment. Where tree stability is already heavily compromised, reduction needs to be sufficient to reduce the crown load center and lever arm to an acceptable level [76].

At this point, it is worth paying attention to the fact that decay in the trees is a normal process (not an illness) and belongs to the life of a tree, and there is and should be decaying parts in old trees. Decaying wood is one of the most important parts of an old tree and is what supports other species and biodiversity. Decay is not always a risk, and a professional must conclude on the basis of VTA and by looking carefully at the structure and studying decay with these non-invasive methods or with microdrill resistance (which may be required in studying large branches in the canopy, where the use of a tomograph is very difficult) whether there are structural weaknesses and what to do about them.

*3.3. Chlorophyll Fluorescence Results*

The results of the conducted research are the results of chlorophyll fluorescence measurements presented with appropriate coefficients (Table 3). The $F_V/F_M$ determinant, commonly used in assessing the impact of stress conditions on plants, is considered a reliable diagnostic indicator. The $F_V/F_M$ ratio not lower than 0.83–0.85 relative units is the optimal value for most vascular plants, indicating their good condition. The influence of various stress factors, both biotic and abiotic, usually contributes to the reduction of this index. The index values at 0.20–0.30 relative units are considered borderline. The $F_V/F_M$ ratio values were lower than the reference ones, especially in trees with visible damage or infected with pathogens. It may suggest a reduced and less efficient efficiency of the photosynthetic apparatus compared to healthy trees. In a stressful situation, the energy reserves of such trees decrease to such an extent that the condition deteriorates and, as a result, weakens the tree, which in extreme cases leads to its dieback. However, the obtained results did not come close to the lower limit, where photosystem II changes to(PSII)'s structure becomes irreversible.

**Table 3.** Results of the chlorophyll fluorescence of trees of senile size in Sandomierz (by authors).

| No. | Species Name | The Obtained Values of the Chlorophyll Fluorescence Index ($F_V/F_M$) in Successive Replications | | | | | Average Value of the $F_V/F_M$ Ratio | Place of Growth |
|---|---|---|---|---|---|---|---|---|
| 1 | Norway maple (*Acer platanoides* L.) | 0.825 | 0.825 | 0.821 | 0.826 | 0.822 | 0.823 | Piszczele Park |
| 2 | White poplar (*Populus alba* L.) | 0.813 | 0.818 | 0.815 | 0.815 | 0.803 | 0.812 | Piszczele Park |
| 3 | White poplar (*Populus alba* L.) | 0.811 | 0.804 | 0.807 | 0.811 | 0.803 | 0.807 | Piszczele Park |
| 4 | English oak (*Quercus robur* L.) | 0.821 | 0.823 | 0.816 | 0.81 | 0.826 | 0.819 | Saski Park |
| 5 | Small-leaved linden (*Tilia cordata* Mill.) | 0.804 | 0.805 | 0.809 | 0.800 | 0.808 | 0.805 | At the entrance to the vineyard and the Church of St. James |
| 6 | Small-leaved linden (*Tilia cordata* Mill.) | 0.779 | 0.755 | 0.708 | 0.769 | 0.780 | 0.758 | At the entrance to the Church of St. James |
| 7 | Common ash (*Fraxinus excelsior* L.) | 0.825 | 0.835 | 0.829 | 0.821 | 0.822 | 0.826 | At the church of St. Paul |
| 8 | English oak (*Quercus robur* L.) | 0.827 | 0.839 | 0.833 | 0.832 | 0.836 | 0.833 | At the rectory of the Church of St. Paul |
| 9 | Small-leaved linden (*Tilia cordata* Mill.) | 0.804 | 0.802 | 0.783 | 0.789 | 0.798 | 0.795 | At the Collegium Gostomianum |
| 10 | English oak (*Quercus robur* L.) | 0.759 | 0.729 | 0.754 | 0.722 | 0.740 | 0.740 | Milberta St. |
| 11 | White poplar (*Populus alba* L.) | 0.821 | 0.815 | 0.821 | 0.820 | 0.822 | 0.819 | Baczyńskiego St. |
| 12 | White poplar (*Populus alba* L.) | 0.807 | 0.808 | 0.818 | 0.809 | 0.829 | 0.814 | Baczyńskiego St. |
| 13 | English oak (*Quercus robur* L.) | 0.824 | 0.826 | 0.826 | 0.830 | 0.826 | 0.826 | Opatowska St. |

The lowest $F_V/F_M$ indexes determining the physiological condition of trees were recorded in the pedunculate oak (inv. No. 10) growing at Milbert Street (0.740), the small-leaved lime growing at Collegium Gostomianum (0.795) (inv. No. 9) and the entrance to the Vineyard of the Dominican Monastery (inv. No. 5) (0.805), and the white poplar (inv. No. 3) from Park Piszczele (0.807) (Table 3). This indicates that the mentioned trees grow under stressful conditions or are subjected to factors that cause them. Their condition is not entirely satisfactory. In the case of oak No. 10, a large (over 30%) branch deadwood, there is leaf infection by powdery oak mildew, the progressive decay of the inside of the trunk caused by yellow sulfur mucus, as well as numerous deep open and surface cavities in which wood decay increases stress, which is reflected in the health of the tree. Among all the examined specimens, this one is the weakest in terms of physiology.

In the small-leaved lime (inv. No. 9) at Collegium Gostomianum, its condition is also significantly impaired due to a massive open loss of the trunk and deep damage to the limbs. It is also caused by a fungal infection visible superficially on some fragments of the boughs and the weakening of the main stem, which was tied several decades ago with a rigid through the bond. The numerous deadwood branches also indicate the weakened vitality of the tree from the physiological point of view.

Lime No. 5, growing in extreme habitat conditions (pressed into a brick fence), is characterized by a similarly weakened vitality. The deterioration of the physiological condition of the tree may also be influenced by a significant open loss of a trunk with a rotting interior and weaker nutrition of the crown, which results in smaller chlorotic leaves, especially in the upper parts of the crown, and 20% deadwood.

Among trees with visible damage, the white poplar (inv. No. 3) from Park Piszczele is in poor condition, especially the trunk. In this tree's case, its physiological state is undoubtedly influenced by the significant activity of saprophytic and parasitic fungi, including peat and sulfurous gall. They contribute to the successive and slow decomposition of the wood, weakening the tree's statics.

Reasonably good health characterizes the remaining stand despite various types of damage to some trees, such as broken branches or boughs, cavities, deadwood, or the activity of insects and fungi.

Pedunculate oaks with inv. No. 8 and 13 (at Opatowska Street and the presbytery of the church of St. Paul), common maple with inv. No. 1 (in the Piszczele Park), and the ash tree amounted to inv. No. 7 (at the church of St. Paul) are in the best condition. Based on the measurements, the chlorophyll fluorescence coefficients in the leaves of these trees were, respectively, 0.833; 0.826; 0.823, and 0.826 relative units, corresponding to a value close to the optimal one, which guarantees the excellent condition of the plant.

*3.4. Valuation of Trees*

Of all the trees included in the analysis, oaks had the highest financial value, regardless of their health condition. It should be said, however, that the items in the best condition are valued the highest and their value, after considering all factors, exceeds EUR 40,000. Those that are damaged to a different extent are also valuable trees, and their monetary value exceeds the level of EUR 20,000. This is mainly related to their dimensions, age, and location. The remaining trees belonging to the faster-growing species (linden, poplars, maple, and ash) are characterized by a slightly lower value than the majestic oaks. Nevertheless, depending on the state of conservation, species, and place of growth, their monetary value is also not small. It ranges from less than EUR 7000 (Norway maple) to almost EUR 17,000 (small-leaved lime).

Disclosing the economic value of the assessed trees to the public may significantly contribute to a greater awareness of the importance of such trees for the environment, increasing the value of the area on which they grow, as well as appropriate care for their health condition. It can also force decision-makers to use effective management methods to maintain such a valuable tree stand in the best condition for an extended time. Tree value calculations are included in the available inventory table (Table 1).

### 3.5. Practical Conclusions

The aim of the study was to formulate recommendations for senile trees. Considering the above, it should be noted that the vast majority of trees assessed in this study are in good health despite visible defects or old age.

A comparison of the results from the tomography and chlorophyll fluorescence is presented in Table 4. Particularly noteworthy are the common ash (inv. No. 7), pedunculate oak (inv. No. 8) and the pedunculate oak (inv. No. 13). The trees mentioned above are in excellent health, which was shown by tests carried out with the use of sonic and electrical tomography. The inside of the trunk of the studied trees is 99–100%of sound wood, without any symptoms of damage or fungal infections. Their physiological state, as confirmed by the fluorescence of chlorophyll, is satisfactory. The obtained results are close to the optimal values for trees growing in good, stress-free habitat conditions.

**Table 4.** A comparison of the results from the tomography and chlorophyll fluorescence of senile trees in Sandomierz (by authors).

| No. | Species Name | Picus Sonic Tomograph 3 | | Picus TreeTronic | Fluorometer OS5p+ |
|---|---|---|---|---|---|
| | | Type of Wood | Percent | | Average Value of the $F_V/F_M$ Ratio |
| 1 | Norway maple (*Acer platanoides* L.) | technically efficient wood | 100% | 56 Ω·m–362 Ω·m | 0.823 |
| 2 | White poplar (*Populus alba* L.) | damaged wood | 63% | 63 Ω·m | 0.812 |
| | | transitional wood | 7% | 1860 Ω·m | |
| | | technically efficient wood | 30% | 2200 Ω·m | |
| 3 | White poplar (*Populus alba* L.) | damaged wood | 56% | 11–33Ω·m | 0.807 |
| | | transitional wood | 13% | 2790 Ω·m | |
| | | technically efficient wood | 31% | 9780 Ω·m | |
| 4 | English oak (*Quercus robur* L.) | damaged wood | 12% | 64 Ω·m | 0.819 |
| | | transitional wood | 4% | 725 Ω·m | |
| | | technically efficient wood | 84% | 2360 Ω·m; resistance values ranged from in the middle of the trunk from 64 Ω m to 206–210 Ω m in the area facing the outside of the trunk (wood typical of pedunculate oak) | |
| 5 | Small-leaved linden (*Tilia cordata* Mill.) | damaged wood | 43% | 7–41 Ω·m | 0.805 |
| | | transitional wood | 24% | 531 Ω·m | |
| | | technically efficient wood | 33% | 2274–2554 Ω·m | |
| 6 | Small-leaved linden (*Tilia cordata* Mill.) | damaged wood | 66% | 75–112 Ω·m | 0.758 |
| | | transitional wood | 8% | 636 Ω·m | |
| | | technically efficient wood | 26% | 4681Ω·m | |
| 7 | Common ash (*Fraxinus excelsior* L.) | damaged wood | 0.7% | 41 Ω·m | 0.826 |
| | | transitional wood | 0.3% | 100 Ω·m | |
| | | technically efficient wood | 99% | 251–262 Ω·m;resistance values ranged from in the middle of the trunk cross-section is a ring system (28–39 Ω m)—suggesting good tissue hydration | |
| 8 | English oak (*Quercus robur* L.) | damaged wood | 0% | - | 0.833 |
| | | transitional wood | 0% | - | |
| | | technically efficient wood | 100% | 298–462 Ω·m;resistance values ranged from 61 Ω m in the middle of the trunk to 153–175 Ω m—may suggest good tissue hydration. | |
| | | transitional wood | 21% | 17,740 Ω·m | |
| | | technically efficient wood | 44% | 15–94 Ω·m | |
| 10 | English oak (*Quercus robur* L.) | damaged wood | 73% | 41–65 Ω·m | 0.740 |
| | | transitional wood | 7% | 220 Ω·m | |
| | | technically efficient wood | 20% | 1171 Ω·m | |

**Table 4.** *Cont.*

| No. | Species Name | Picus Sonic Tomograph 3 | | Picus TreeTronic | Fluorometer OS5p+ |
|---|---|---|---|---|---|
| | | Type of Wood | Percent | | Average Value of the $F_V/F_M$ Ratio |
| 11 | White poplar (*Populus alba* L.) | damaged wood | 24% | 80–86 Ω·m | 0.819 |
| | | transitional wood | 20% | 200 Ω·m | |
| | | technically efficient wood | 56% | 700 Ω·m | |
| 12 | White poplar (*Populus alba* L.) | damaged wood | 23% | 122 516Ω·m | 0.814 |
| | | transitional wood | 15% | 9500 Ω·m | |
| | | technically efficient wood | 62% | 8000 Ω·m | |
| 13 | English oak (*Quercus robur* L.) | damaged wood | 0% | - | 0.826 |
| | | transitional wood | 0% | - | |
| | | technically efficient wood | 100% | 338 to 414 Ω·m; resistance values ranged from 90 Ω m in the middle of the trunk to 129 Ω m, which is caused by good tissue hydration. | |

The most problematic trees in terms of their health and physiological condition are the English oak (inv. No. 10) growing at Milbert Street, the white poplar from Park Piszczele (inv. No. 3), and the two lime trees—one at the entrance to the Dominican Vineyard (inv. No. 5) and the other at Collegium Gostomianum (inv. No. 9). These trees have significant deep cavities, are infected with fungal pathogens, and are characterized by a low $F_V/F_M$ ratio determining the physiological state, which indicates growth under stressful conditions. The oak growing at Milbert Street has significant deep cavities and considerable damage, and the infection by sulfuric gallbladder does not give any optimism about its future. However, it is worth treating this tree with respect and allowing it to grow as long as its vital forces allow it. It is a tree historically associated with the area in which it grows; thus, attempting to minimize the impact of unfavorable factors will allow it to survive for a long time. In the case of the white poplar from Piszczele Park, one should consider the progressive decomposition of the inside of the trunk caused by the sulfuric gallbladder. The tree is still relatively stable due to the produced column that supports and strengthens the trunk from the damaged side. Nevertheless, regular monitoring will allow appropriate steps for its continued existence in the future. The two lime trees mentioned above are already senile and strongly "experienced by life". The rigid pass-through bonds strengthen their trunks, reducing the potential risk of breaking them. The vitality of trees is still significant, which allows us to think positively about the prognosis for the future.

The rest of the assessed trees are in different health conditions but are good enough that, currently, they require only minor maintenance.

Old trees are a precious component of tall greenery for the city, and even despite their advanced age or considerable damage, they are an invaluable element of biodiversity. Each too-hasty decision to remove a tree is a massive loss for the environment, both in terms of image, nature, and history, as well as finances, which is not always remembered.

The management of veteran trees is in the first place improving the safety and the living conditions around the tree and, as last options, touching the tree (e.g., pruning).

The area on which senile trees grow should be equally adequately protected, and, in the event of an excessive threat from them, appropriate steps should be taken to minimize the risk to the environment. Considering the so-called tree stand management, it is advisable to carry out appropriate treatments to keep the trees in reasonably good condition (Table 5).Where soil conditions under a veteran tree appear to be unfavorable—for example, severe soil compaction—the first thing to do is to stop or reduce the causes of those conditions. In the case of trees in Sandomierz, it is suggested to scatter the bark under trees No. 1, 2, and 3 and systematically mow the lawn grass under the others.

**Table 5.** Management plan forsenile treesin Sandomierz (by authors).

| No. | Species Name | Removal of Branches | Crown Correction (Removal of Stumps) | Health Condition Monitoring | Ordering the Tree Surrounding | Other |
|---|---|---|---|---|---|---|
| 1 | Norway maple (*Acer platanoides* L.) | yes (branch deadwood 25%) | no | yes (every two years) | yes | - |
| 2 | White poplar (*Populus alba* L.) | yes (branch deadwood 10–15%) | no | yes (each year) | no | - |
| 3 | White poplar (*Populus alba* L.) | yes (branch deadwood 15%) | no | yes (every six months) | no | - |
| 4 | English oak (*Quercus robur* L.) | yes (branch deadwood 25%) | no | yes (each year) | no | - |
| 5 | Small-leaved linden (*Tilia cordata* Mill.) | yes (branch deadwood 20%) | no | yes (each year) | no | - |
| 6 | Small-leaved linden (*Tilia cordata* Mill.) | no (branch deadwood 10–15%) | no | yes (each year) | no | - |
| 7 | Common ash (*Fraxinus excelsior* L.) | yes (branch deadwood 20%) | no | yes (every two years) | no | - |
| 8 | English oak (*Quercus robur* L.) | yes (branch deadwood 20%) | no | yes (every two years) | no | - |
| 9 | Small-leaved linden (*Tilia cordata* Mill.) | yes (branch deadwood 15%) | no | yes (each year) | no | - |
| 10 | English oak (*Quercus robur* L.) | yes (branch deadwood 30%) | yes | yes (each year) | yes | - |
| 11 | White poplar (*Populus alba* L.) | yes (branch deadwood 10–15%) | yes | yes (each year) | no | - |
| 12 | White poplar (*Populus alba* L.) | yes (branch deadwood 10–15%) | yes | yes (each year) | no | - |
| 13 | English oak (*Quercus robur* L.) | yes (branch deadwood 5%) | no | yes (every two years) | no | Protected by an entry in the register of monuments |

Tree species do differ in their ecological requirements for light, such as for germination and establishment, and in their ability to tolerate some shade [4,5,77]. However, all trees need light, especially trees in, or entering, the senile stage of life. In terms of trees' access to light, no problems were noticed. Only tree No. 1 is located in a thicket of neighboring trees, therefore, bushes clearing is suggested.

## 4. Discussion—Combination of Four Tree Assessment Methods

The above results confirm that the use of non-invasive methods is a source of practical information about the health condition of trees. The conducted investigations and analyses, as well as the application of the obtained results, can be used to protect senile trees. The

presented results are a response to the need for research on trees of monumental size in the field of sustainable urban greenery management and the perception of the economic value of trees by the community. Old trees growing in urban areas are an essentially natural and cultural element but require unique solutions to minimize risk in their surroundings. The survival, growth, and management of street trees in a city pose unique problems, as most street trees grow in a stressful city environment. The negative urban factors affecting trees are soil compaction, variable water-air ratios, water shortages due to surface runoff, high temperature, salt aerosol, or air pollution. High air temperature and its often bad composition ($CO_2$, $O_3$, $NO_2$, etc.) are unfavorable for plants, especially those growing on paved areas, such as squares. The mentioned factors influence the physiology of trees, e.g., photosynthesis or enzymatic activity [78].In addition, poor air quality can cause mechanical changes such as obstruction of the stomata caused by dust suspended in polluted air. Such changes result in the deterioration of gas exchange; thus, they disturb the life processes of trees [79]. Climate change effects, such as heavy rainfall and local tornadoes, also increase the risk of tree damage [80]. Moreover, studies also indicate that communication with the public is an obstacle to effective management, as the public is not sufficiently aware of the benefits of trees [81,82]. Many inhabitants do not perceive the importance of street trees due to unknown economic values [83,84].

The fall of a tree or limb can cause significant damage to public infrastructure, personal property, and even human life [85], and, when discussing research on the biomechanics of trees, it is worth noting the difference between forest trees and trees growing individually in an open field. The latter group includes most of the natural monuments in the Sandomierz. Open-grown trees usually grow with considerable branch mass, and the dynamic response in winds may be different to other tree forms. The shape or morphology of the tree and the distribution of oscillating branch masses become important during dynamic studies [86]. For example, slender forest conifers sway in a relatively simple manner in contrast to open-grown trees, which have many independent and larger branch masses [87]. The dynamic interaction of branches in winds can significantly modify the frequency and damping of a tree [88]. Studies examining tree failure due to winds in urban areas have been undertaken after wind storms, but with only limited correlation to actual wind velocity and gustiness [89,90].There is currently no definitive method to predict the failure of an individual tree. Assessments of trees include visual tree assessment [91], tree risk assessment methodology [92], quantified tree risk assessment [93], and static integrated methods that combine static pulling with dynamic wind load assessment [94]. It is noteworthy that crown thinning was less effective at reducing trunk movement [95]. Branches on thinned trees appeared to move more than branches on other treatments but not in the same direction. This complex branch movement indicates that the dynamic effects of branches may play an important role in acting as a buffer to dampen and reduce motion [88].

Urban trees, distinct from forest trees, are characterized by complex growth environments, a greater vulnerability to damage, and wind tunnel effects caused by high-rise construction [96–98]. Therefore, the arborist control of old trees is essential in the city. It requires precise diagnostic techniques to detect, for example, rot, internal decay, or other structural defects in tree trunks. Visual tree assessment (VTA) is still the starting point for such research. However, internal defects of tree trunks often remain beyond the sight of a specialist [99–101]. However, there are technological tools to assist with tree inspection. These tools provide results that can be used in calculating fall risk, contributing to the decision to keep or remove the tree.

The most appropriate methods have minimal effect on living tissues, e.g., acoustic tomography, giving 80% accuracy of the results [102,103]. Over the past two decades, technologies for non-invasive research and the evaluation of urban trees have attracted the attention of dendrologists worldwide [104–106]. It is interesting, for example, to compare sound tomography and GPR surveys. Research using non-destructive radar techniques (GPR) can be a diagnostic tool to assess the health status of living tree trunks based on the

internal distribution of dielectric permittivity. However, it is still a method that requires refinement and is very time-consuming (some hardware modifications are needed to examine the tree). Compared to tomographic diagnostics, it also does not provide easy-to-interpret images and requires, at the same time, extensive experience in reading the obtained results [107,108]. On the other hand, acoustic and electrical tomography is a perfect combination. While acoustic techniques provide detailed information on the quality of the wood, they are ineffective when it comes to differences between decayed wood and bacterial wet wood or decay and wood loss. In contrast, electrical resistivity methods can identify wood decay early.

However, it is worth noting that the resistivity tomography method provides low spatial resolution. The first pioneering works on the electrical resistivity of wood in living trees were published in 1965–1995. The best-known example of an electrical conductivity meter is the Shigometer, which consists of a twisted wire probe and resistivity meter [109]. When operating the device, an electrical probe is placed into a small, pre-drilled hole approximately 3 mm in diameter. The pattern of resistivity of the wood to a pulsed direct current is recorded [110]. Shigo (1991) claimed that, in the region adjacent to wood decay, the concentration of cations in the wood would increase and, therefore, the electrical resistivity would decrease. However, electrical resistivity also decreases if wood is healthy but dry [111] and increases in dry decayed wood and when the probe moves from sapwood to heartwood [112]. Electrical resistivity may also decrease when the probe reaches bacterial wetwood [109]. Several researchers have had very inconsistent results with the Shigometer. Results have been dependent on the tightness of fit of the electrode and the relative moisture and resin content of the timber [110]. Shigo and Shortle (1985) advised that the Shigometer does not function in resin soaked, frozen, or dead wood, and the operator should carefully control the amount of contact the needle electrodes have with the wood [109]. Nicolotti et al. (2003) assessed results from electrical resistivity and ultrasonic tomography and georadar [113]. They reported good results with electric tomography. Electrical tomography was deemed promising by Hagrey (2007), but the results were qualitative rather than quantitative [114]. Problems with the Shigometer in eucalypts may be similar to electrical tomography, as the raw data are the same (electrical resistivity).

The test carried out in the Sandomierz showed that the measurement should be performed with sonic tomography, and then the result should be confirmed with an electric tomograph. The tomogram obtained from the sonic tomograph may show the inside of the trunk in blue, but we do not know if it is void or rot. On the other hand, an electric tomograph confirms one of these variants.

Two or more methods are usually combined to examine the health status of a tree for comparison and the validation of the results [115,116]. In this case, in Sandomierz, additional studies of chlorophyll fluorescence and the economic valuation of trees were carried out to support the effective monitoring and management of trees of considerable size.

The EU ecological policy formulates postulates on the knowledge, inventory, valuation, and protection of ecosystem services. Provisions in this regard are contained in the EU biodiversity strategy [117] and the Seventh Environmental Action Program [118]. In urbanized areas with a small share of natural ecosystems, trees play a unique role in shaping the environment of human life. The role of trees in urban space has been widely discussed and documented in the achievements of science. They provide a range of natural and cultural benefits. At the same time, the awareness of the authorities, administration, and residents about the trees found in a given area, their importance, and their economic value is often limited. Trees are treated as one of the prominent elements of the urban landscape. There is not much attention paid to the role they play in shaping the quality of life in the city.

For many years, economists were only interested in wood's very narrowly understood utility value. Unfortunately, in Poland, the practical approach to nature and the emphasis on its servant role towards humans are still dominant, as well as the perception of activities for the protection of nature and its preservation as obstacles to the socio-economic development

of a commune, region, or country. With this perception of nature, a tree growing outside the forest, e.g., in a city, is—in the opinion of many property owners—primarily an obstacle in the development and use of a given area, a source of nuisance in its maintenance, or a threat to the environment, and the value of the tree is determined only by the value of wood cutting [36].

It is worth noting that most tree ecosystem services can also be estimated and the value can be expressed in the form of a specific economic value. One of the most famous examples of the use of tree ecosystem services valuation recently has been a study conducted in New York. Their subject was evaluating costs and benefits related to street trees managed by the city authorities. The study covered almost 600 thousand such trees (but 4.5 million trees growing in parks and private lands were not included in it). The analyzed trees yielded net benefits (after adjusting for costs) of USD122 million per year (USD 209 per tree); for every dollar spent on tree maintenance in New York City, the city benefits USD 5.6. Benefits such as the reduction of energy consumption, the absorption of $CO_2$ and other pollutants, water retention in the landscape, or the impact on the property value were considered [119]. Experience in evaluating various natural resources in cities on a global scale includes tens, perhaps hundreds of thousands of such studies, but in Poland, such studies have so far been carried out rather sporadically.

Preparing the valuation of trees and including them, e.g., in analyses of the value of the property, should first become a recommended, and ultimately even mandatory, tool for environmental management, as well as property management, especially by municipal governments and other entities managing public space, including the State Treasury [120].

The example of activities presented in the manuscript shows the possibilities of implementing the postulates of ecological policy by local authorities, guided by the need for in-depth economic knowledge about the local natural environment and improving the quality of life of inhabitants of urbanized areas. The economic valuation of trees facilitates the decision-making process regarding tree stand management in urban areas. It allows us to reduce common denominator issues that are often difficult to decide due to the diverse needs, perspectives, and positions of society presented, expressing their value in money, which continually facilitates the message about the value of things and places, including those seemingly immeasurable.

## 5. Conclusions

This study presented a combination of several non-invasive technologies for detecting and assessing the health status of urban trees. The combination of visual inspection (VTA), acoustic and electrical tomography, and stress testing with a fluorometer is a practical approach to measuring the health of various tree species. The procedures developed by us enabled the analysis of the conservation status of endangered trees and may improve the risk assessment process caused by them.

Non-invasive tools using acoustics and electric waves are still new in Poland, but the authors of this manuscript conducted similar research (one of the first in Poland) in the case of the historic greenery of the Botanical Garden in Lublin. Then, the effectiveness of the use of tomographic tests in the management of greenery in historic gardens was confirmed. The research presented in this manuscript was supplemented with fluorimetry and tree valuation results. The valuation of a tree is an additional protection tool because it contributes to its public approval and sometimes makes the inhabitants aware of the natural value of an old, often problematic specimen. Thanks to the use of combined methods of tree assessment, it is possible to diagnose old specimens, take care of the safety of people and property in their vicinity, and test candidates qualified for legal protection. Combined methods of tree assessment can be an important activity in the sustainable management of urban greenery. This case study of 13 trees growing in Sandomierz contributes to broadening the knowledge about this technique and its application in Poland and Europe.

**Supplementary Materials:** The following supporting information can be downloaded at: https://www.mdpi.com/article/10.3390/land11111914/s1, Supplementary File S1.

**Author Contributions:** Conceptualization, M.D., W.D. and M.M.; methodology, M.D. and W.D.; software W.D., formal analysis.; investigation, M.D. and W.D.; data curation, M.D. and W.D.; writing—original draft preparation M.D. and W.D.; writing—review and editing, M.D., W.D. and M.M.; visualization, M.D. and W.D.; supervision, M.D. and W.D. All authors have read and agreed to the published version of the manuscript.

**Funding:** This research received no external funding.

**Data Availability Statement:** Not applicable.

**Acknowledgments:** The research was financed from the own funds of the Department of Landscape Architecture and the Institute of Horticultural Production.

**Conflicts of Interest:** The authors declare no conflict of interest.

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
