# Peer review of "A Combined Methods of Senile Trees Inventory in Sustainable Urban Greenery Management on the Example of the City of Sandomierz (Poland)"

_land, doi:10.3390/land11111914_

Round 1

Reviewer 1 Report

This interesting manuscript deals with an important theme in the urban environment: management of veteran trees, which are very valuable for biodiversity, human well-being, cultural history and for many other reasons. Studying the condition of the trees is the first step to make a management plan to preserve these trees safely in the environment. The manuscrip studies different methods in evaluating the condition of the trees, and as well as a method for calculating the tree replacement value.

I recommend to reconsider the publication after major revision. My further comments are in the file, attached. In general, the most important revisions: Materials and methods are missing some important info about how the measurements were done. As well, the valuation method of trees needs more  clarification. On the other hand, this section can be condenced. The same applies to the section 3.2., which must be shortened and condenced. Tha authors are adviced to consider using more tables in presenting the results.

I encourage the authors to work on the manuscript and to resubmit it after major revision. I find that this manuscript and this theme would be valuable for publishing in an improved form.

Author Response

Dear Sir/ Madame. Thank you kindly for all comments. Thanks to his suggestions, I was able to try to improve this manuscript. Thank you for your time and insightful opinion on the article. Yours faithfully, Authors.

Reviewer 2 Report

The article is not very precise,

More citations from international literature needed.

It is necessary to describe and discuss the characteristics of a senile tree

Holow tree and cavity is not a disease - it is typical trait of the ageing phase

There needs to be a discussion about hollowness and the effect on the statics of senile trees, the role of hollowness

L 50 and assessing the health condition of city trees- rather the resistance of the trunk to fracture - this is not an assessment of the condition but of the extent of cavity  

L 53 the health status  - evaluation of the statics and condition of trees (two different but both important issues)

L 49 – 56 – belongs to Mathodology section

80-96- to long

L 101 in that place thesis should be formulated

point 1.2 belongs to M&M section

l 164 the age of the tree is needed? Maybe assessment if its young, mature or senescent/ancient is more specific?

175 the number of  incidents and financial damage caused by falling trees in cities is increasing – source of information is needed

178 of tree condition assessment – VTA is the method of risk assessment

183 resistor – you mean resistograph? Schwarze in a book proved non invasive tool of resistograph investigation

How harmful is testing with resistograph?

Point 3.2 its not discussion – it belongs to the result section

L 324 that there is no decomposition inside the trunk only - the CT scan does not tell you about the health of the tree

L 349 we don't know if it will progress - especially in the case of mature trees we know the extent of it and can check in time if it is progressing 

L 372 will undoubtedly worsen over time – we don't know that either

Point 3.3 mixes condition with signs of decay, a typical feature of aged trees It is wrong and needs to be rewritten Condition and statics affected by decay are two different things.

L 668 is 99-100% healthy??? Has a sound wood

L 708-734 that is introduction section

The discussion and conclusions section must be rewritten after the results have been sorted out

Author Response

(The authors gave the same response as above.)

Reviewer 3 Report

Focus seems to be on non-invasive assessment of the monument trees and leads to recommendations for monitoring and management. I think portions of this can be re-written to focus on that and the details of each tree could be a supplemental file for the interested reader.

The title seems awkward - a hybrid model but nothing is modelled. Maybe say something about non-invasive assessment of health conditions?

Think authors should summarize the findings of the trees - for example, indicated by tomography, a cavity here indicates a health issue. Maybe create a table to show the percentage of each tree with tissue damage, etc.

Lines 22-24: "According to the authors of this article..." This reads awkwardly and could be rewritten to provide the information on the results leading to actionable recommendations.

Line 41: "Old trees..." Think this line can be removed.

Line 65: Ancient trees are precious for a...(instead of the)

Lines 66-70: Rework into two sentences.

Line 86-87 have redundant wording

Lines 89-90: "Different people..." There are some typos in this sentence.

Line 90: "The owner..." Change to The owners of a ...

Lines 108-109: Say 13 trees, here 12...might point out here since it is introduced that one tree is a candidate tree?

Lines 204-205: Think this is an example of something that would be useful in the results. This information on ranges in the speed of sound would help explain those figures later.

Lines 251-252: Where - at what level in the crown - were the clips placed? Would that matter for amount of sunlight received in different portions of the tree crowns?

Lines 289-298: Where does this information come from? Seems this (economic analysis) is out of place in a paper focused on health assessment using tomography, etc.

Line 314: Discussion? Is this results of tomography? Discussion seems out of place here

Line 323: indicates is a better word than prove - you may need something more invasive to prove/verify moisture presence. This word needs changed throughout.

Line 336: An example where referencing a figure and color changes/differences would help.

Table 2: Consider making these figures in the Results that detail what is being seen in the images - wet wood, decay, etc.

Lines 645-660: Recommend removing this section/economic portion of study.

Lines 799-802: Can the authors provide a citation to support this claim?

Author Response

(The authors gave the same response as above.)

Reviewer 4 Report

Manuscript ID: land-1946781

Title: A hybrid model of trees of monumental size inventory in sustainable urban greenery management on the example of the city of Sandomierz (Poland)

The topic is very interesting, also the data collected by Authors is valuable corresponds very well to the scope and aims of the journal. However, the manuscript is very long and in chaotic form present divers scope of aspects related to old selected trees as case studies, the main aim is not clear. The presentation of selected parts has some weaknesses which decrease the scientific value of conducted study – the order of presented parts needs more methodological approach. The writing (style) suggests that it is more a sort of a ‘story’, than a research paper. Thus the manuscript should be improved.

Main comments and suggestions for Authors are listed below:

1. Abstract is rather long but at the same time not well organized, the main objective of the study is not presented. In the second part of the Abstract the description sounds more as the Authors' discussion of what the research could bring, than the main results. Therefore, the Abstract must be improved.

Key words are generally well selected, but they do not imply any trees - they should be also developed in that direction, in my opinion.

2. The main objective of research is not formulated – in lines 49-51 (in the middle part of Introduction) - Authors just say that ‘The research ... presents a combination of several non-invasive methods for detecting damage and cavities inside trunks and assessing the health condition of city trees.’ without clear arguments why it is important to use these few methods. It is not clear if the general idea of the study is to present methods, their results or values in the context of trees health status valuation, to compare results of few methods, or to count the value of trees in general, or to formulate the management recommendations, etc. Regarding so many aspects presented in the manuscript the  aim must be very clearly presented, after the introductory part.

3. Introduction

- in some parts of section 1.1 there are incomprehensible references, e.g. Authors say that 'In North America, it is estimated that urban trees live on 59 average for about 20 years.' (lines 49-50), while the study is conducted in limited/specific area (city) of one European country. The aspects presented in subsections of Introduction must be directly related to the specific of the study area and/or region, because the study is not a review/comparison of others.

- some phrases are not well used, for example 'ancient trees' (line 65) - while the subject is rather old trees/monumental trees, etc. At the same time, naming trees as 'objects' or 'research objects' (line 102) is a bit artificial/confusing. I suggest to call them consequently trees (or plants) - the nomenclature used should be precise and consistently repeated throughout the manuscript.

5. Material and Methods

- the part related to the main information of trees proposed for research (cases and their description) presented in subsection 1.2. of Introduction is a part of  ‘Material’, thus should be moved to the section of Material and Methods. I also suggest to rewrite this part - it would be professional to start with short explanation why these trees were selected for the study and also that they grow in various spatial situations: parks, sacred areas, …., etc. The main general data of trees should be presented in a newly created table, including their names, N-E location (this data should be removed from Table 1), and type of location, etc.

- the presentation of methods used for the research must be also very clear – the present form of their description has a continuous form what is difficult to read and understand. The presentation of each method should follow the same form, e.g. name of the method, type of aspect which are related to the method, a form (way) of not invasive measurement - way of implementation, role of the method for the tree assessment, etc., and should be supported by the literature. I suggest to divide this part of manuscript into subsections with individual numbers of each method to make it more easy to understand.

- the presentation of the method of tree valuation (in lines 276-298) contains a mathematical formula, but there is no detailed data and/or explanation of its individual parameters, thus it is not clear what data and taken from which sources are finally used. For example, is the ‘actual value of a tree with a trunk circumference of more than 25 cm’ or ‘Value (Euro)’ in Table 2 taken from some law acts or a valuation indicators used in Poland, or any other international rules? it must be explain. The lack of attached sources/data such as MydÅ‚owska's tables or Majdecki’s tables make the use of this data no clear. The rule must be more explained.

6. Results – this section needs to use more clear the main form and order of the results presentation. The continuous form of description is very long and sounds as a ‘story’. I suggest to limit the description of each tree in favor of collecting the main data in the table including columns dedicated to each method, also presentation of observed destructions of each tree should be added in a short way in separate column of this summary table – it is still not clear if the idea of research is to present results of each method separately, to compare them in relation to each tree, and finally evaluate the trees between them, or something else. The presentation of results must well argued and improved.

- the title of subsection 3.2. Discussion is incorrect, this description is not a Discussion.

7. It is not clear what means the formulation ‘practical conclusions? (line 662). The description presented in this section is generally not clear - Authors propose some management recommendations for each tree but they do not apply much to the obtained results – it is still not clear if the aim of the study was to formulate recommendations for old trees or something else… Also, if any recommendations are possible to be formulated, they should be a part of Conclusions, but rather related to the use of diverse methods to evaluate trees than detailed maintenance treatments.

8. Discussion – the scope of discussion focus mostly on two aspects: the value of presented methods more than results obtained, and also some general relations of the values of trees as a part of ecosystem services, etc. In my opinion, the discussion must directly follow the role of used not invasive methods as a source of practical information about the health condition of trees, and application of their results for the protection, than repeating general statements that trees are an important element of ecosystem services, because this is not the result of the conducted study.

Others:

The English language should be improved, this suggestion is applicable to the use of selected formulations listed above, but also to some repeated words in following sentences and/or in one sentence such as: ‘The research presented in this article presents a’ (line 49), etc. – generally the selection of words should be more balanced. Also some sentences have not good grammatical order.

Showing measuring devices - e.g. on Figures (Fig. 7,8) - is unnecessary, it looks like an advertisement, when the key is the selection and validity of each method and the obtained results, etc.

Summing up, the manuscript needs major revision in almost all parts, thus I can not recommend it to be published in the present form.

Author Response

(The authors gave the same response as above.)

Round 2

Reviewer 2 Report

Dear Authors, 

thank you for improuving the paper, it could be developed further but in present form is acceptable 

Best Regards 

Author Response

Dear Sir/Madame. 
Thank you very much for reviewing the revised version of manuscript. Thank you for your time and second opinion on the article. Yours faithfully, author.

Reviewer 3 Report

There are still a few sentences from the previous review that I think should be reworded.

The results of individual trees, if included, should be an appendix for the interested reader. If the focus here is on tomography - the method and results/insights obtained, that should be the focus of the results. I think the figures showing the tomography images are great and help tell the story the authors are trying to tell. On the same lines, the economic analysis has no bearing on the current manuscript and I recommend removing it.

Author Response

Dear Sir/Madame. 
Thank you very much for reviewing the revised version of manuscript. Thanks to your suggestions, I was able to try to improve this article. Thank you for your time and second opinion on the article. Yours faithfully, author

Reviewer 4 Report

Manuscript ID: land-1946781

I appreciate all works done by Authors. The improved order of main elements of the manuscript makes its structure more clear, also the presentation of the study represents better methodological approach. The new title fits better to the scope of the study. The Abstract is acceptable, also key words. Introduction section presents important aspects of the study, but the scope of references is not very wide. The methods’ presentation is long but quite clear after corrections.

Last important Comments and Suggestions for Authors:

1. The part of the main aim of the study is unfortunately still presented at the end of the manuscript – in line 817. This most important sentence informing about the main aim must be absolutely moved to the introductory section, to be very easy to find by readers, thus I suggest to add information that ‘The objective of the study is to formulate recommendations for senile trees based on the results of their condition assessment obtained with the use of three non-invasive methods’, etc.

2. Results, including the improvement of tables, are generally more clear and easy to understand, even if the description is very long... Regarding the scope of improvements implemented to the manuscript, the management plan presented in Table 5 is however very general in its present form, especially in the case when most of treatments are obvious such as ‘Health condition monitoring’ (applicable for all trees), and especially the information in column 7 ‘Protected by an entry in the register of monuments’ sounds strange and means that 12 of studied trees should not be entered in the registry as they are already there... Especially this information should be included in the text, not in the table, to avoid chaos. Also the formulation ‘Ordering the environment’ means rather ‘Ordering the tree (close) surrounding’. In my opinion this table doesn’t tell much, and regarding so detailed research on each tree, it would be necessary to define more precisely, for example, the degree of listed treatments, etc. It would be more valuable for the management plan, etc. 

3. Some more references to literature would enhance the scientific value of the discussion.

Others:

- the title of Table 5. is not clear - use the word ‘trees’ just once.

- I suggest to eliminate the term ‘given place’ (e.g. line 168) and to replace it by: location

- the manuscript still needs dome general language correction

Author Response

Dear Sir/Madame. 
Thank you very much for reviewing the revised version of manuscript. Thanks to your suggestions, I was able to try to improve this article. Thank you for your time and insightful second opinion on the article. Yours faithfully, author
